# Development of A Novel Corrugated Polyvinylidene difluoride Membrane via Improved Imprinting Technique for Membrane Distillation

**DOI:** 10.3390/polym11050865

**Published:** 2019-05-13

**Authors:** Normi Izati Mat Nawi, Muhammad Roil Bilad, Nurazrina Zolkhiflee, Nik Abdul Hadi Nordin, Woei Jye Lau, Thanitporn Narkkun, Kajornsak Faungnawakij, Nasrul Arahman, Teuku Meurah Indra Mahlia

**Affiliations:** 1Chemical Engineering Department, Universiti Teknologi PETRONAS, Seri Iskandar, Perak 32610, Malaysia; normi_16000457@utp.edu.my (N.I.M.N.); azrinazolkhiflee@gmail.com (N.Z.); nahadi.sapiaa@utp.edu.my (N.A.H.N.); 2Ad Advanced Membrane Technology Research Centre (AMTEC), Universiti Teknologi Malaysia, Skudai 81310, Johor, Malaysia; wjlau@petroleum.utm.my; 3Nanomaterials for Energy and Catalysis Laboratory, National Nanotechnology Center (NANOTEC), National Science and Technology Development Agency (NSTDA), 111 Thailand Science Park, Pathum Thani 12120, Thailand; thanitporn.nar@ncr.nstda.or.th (T.N.); kajornsak@nanotec.or.th (K.F.); 4Department of Chemical Engineering, Universitas Syiah Kuala, Jl. Syeh A Rauf, 7, Darussalam, Banda Aceh 23111, Indonesia; nasrular@unsyiah.ac.id; 5School of Information, Systems and Modelling, Faculty of Engineering and Information Technology, University of Technology Sydney, NSW 2007, Australia; TMIndra.Mahlia@uts.edu.au

**Keywords:** membrane distillation, corrugated membrane, membrane fouling, membrane wetting, sustainable desalination

## Abstract

Membrane distillation (MD) is an attractive technology for desalination, mainly because its performance that is almost independent of feed solute concentration as opposed to the reverse osmosis process. However, its widespread application is still limited by the low water flux, low wetting resistance and high scaling vulnerability. This study focuses on addressing those limitations by developing a novel corrugated polyvinylidene difluoride (PVDF) membrane via an improved imprinting technique for MD. Corrugations on the membrane surface are designed to offer an effective surface area and at the same time act as a turbulence promoter to induce hydrodynamic by reducing temperature polarization. Results show that imprinting of spacer could help to induce surface corrugation. Pore defect could be minimized by employing a dual layer membrane. In short term run experiment, the corrugated membrane shows a flux of 23.1 Lm^−2^h^−1^ and a salt rejection of >99%, higher than the referenced flat membrane (flux of 18.0 Lm^−2^h^−1^ and similar rejection). The flux advantage can be ascribed by the larger effective surface area of the membrane coupled with larger pore size. The flux advantage could be maintained in the long-term operation of 50 h at a value of 8.6 Lm^−2^h^−1^. However, the flux performance slightly deteriorates over time mainly due to wetting and scaling. An attempt to overcome this limitation should be a focus of the future study, especially by exploring the role of cross-flow velocity in combination with the corrugated surface in inducing local mixing and enhancing system performance.

## 1. Introduction

Freshwater scarcity has become a big challenge worldwide and inevitably would lead to a global crisis if not anticipated properly. This issue emerges since the rapid demand for fresh water is not matched with the supply. Moreover, increasing water consumption driven by a growing population, improved living standards, flourishing agricultural sector and industrialization have played major roles in worsening the issue. Greve et al. [1] also reported that in the coming decades, climate and societal changes are projected to further worsen water scarcity in many regions worldwide.

To cope with this situation, membrane distillation (MD) has been regarded as one of the promising technologies for water desalination to produce fresh water [2,3]. MD is highly attractive for desalting highly saline water [2,4], where established technology such as reverse osmosis (RO) is not economically feasible because the performance of MD is only slightly affected by solute concentration. Moreover, MD can be operated at lower temperatures than other thermally driven methods (i.e., evaporation) that work at the boiling point, thus exhibits operational advantages. In the most basic form of MD for desalination, a hot saline feed solution flows over one side, and cold stream flows on the other side creating water vapor difference between both sides of the membrane. As a result, water liquid evaporates at the hot side, transports across the membrane and condenses at the cool side. Nonetheless, MD is only potentially cost-effective in comparison to RO when paired with readily heat sources [5,6].

MD purifies water using a hydrophobic membrane that is permeable to water vapor while disfavoring liquid water to permeate. Thus, an ideal membrane for MD should possess the following properties: high hydrophobicity, pore size that ranges within micro and ultrafiltration, a narrow distribution of pore size, and high fouling resistance [7,8]. Development of an MD membrane should consider those requirements to result in attractive materials. To date, typical polymers that commercially used as a membrane materials are polytetrafluoroethylene (PTFE), polypropylene (PP) and PVDF. PP is frequently used as a membrane material in the earlier years (1991–2010) due to its high hydrophobicity, good chemical and thermal resistance [8,9]. However, recently (2011–present), there is a remarkable decrease in the use of PP because of its low mechanical stability and due to the reduction of capillary membrane for MD application [8]. Meanwhile, PTFE has the best hydrophobicity, chemical and thermal stable yet it is not the best option for MD due to its high thermal conductivity other than relatively expensive material and too stable for modification and functionalization [10,11]. Thus, PVDF received great attention by researchers since it possesses outstanding properties suitable for MD membrane which are high hydrophobicity, high thermal stability, good chemical resistance and excellent mechanical strength [9,12,13].

Membrane fouling is one of the factors affecting MD performance, especially when treating scaling-prone feeds, i.e., feeds containing Mg or Ca with low solubility. Fouling is simply referred to scaling when the deposit is predominated by inorganics. MD suffers from severe scaling when treating a high salinity feed, especially near solute saturation concentration. Scalant could precipitate on the membrane surface, inside the pores or penetrate across the membrane. Subsequently, fouling/scaling diminishes productivity, promoting wetting and deteriorating permeate quality. In view of this, it demands pre-treatment, control, and system maintenance [14]. As an example, Tijing et al. [15] reported that calcium phosphate scaling severely deteriorates plant performance that eventually inflates operational costs. Liu et al. [16] developed a unique asymmetric PVDF membrane, which specifically designed for MD application with excellent anti-fouling properties. The membrane was fabricated using nonsolvent thermally induced phase separation (NTIPS) method and applied for the concentrating process of semi-product of organic fertilizer.

Recent reports focus on fabrication of a new class of membranes having surface patterns. Table 1 summarizes recent studies on a corrugated membrane for various applications. The patterns have proven effective to mitigate scaling for various membranes processes, including MD. One of the approaches is to form surface corrugations. Corrugation increases effective membrane surface area and acts as a turbulence promoter to suppress fouling [7,17]. Its effectiveness is not only proven via experimental studies but also confirmed via modelling. Usta et al. [18] performed a computational fluid dynamics simulation, under both laminar and turbulence flow, for a membrane containing triangular chevrons and square ribs spacers attached to its surface. The results show that turbulence flow enhances performance by minimizing membrane fouling. Scott et al. [19] also fabricated corrugated membranes for crossflow membrane microfiltration of water-in-oil emulsions. Mechanical pressing between metal dies at 120 °C was applied on the membrane sheet to form corrugation in the membrane surface.

One effective way to form corrugation on the membrane surface is via templating method. This method has recently been reported in the form of the corrugated composite membrane [17]. The membrane consists of two composite layers, in which the first layer serves as a base membrane where the overall pore size is controlled and the second layer (on top of the first layer) as a platform to form surface corrugations and to induce hydrophobicity. Despite its advantages of improving wetting resistance, the membrane suffers from relatively low flux, about half of a flat membrane used as the reference [7]. In this study, we modified the templating method to develop a new corrugated membrane, aiming to achieve both high flux and wetting resistant. The novel aspects of this study are on the location of the corrugated layer to maintain acceptable pore size and higher temperature of the dope solution to suppress the crystallization of semi-crystalline polyvinylidene difluoride (PVDF). After preparation of both flat and corrugated membrane, they were characterized. Their hydraulic performance was later evaluated in short-term MD test treating both seawater and brine feeds. Their wetting resistance was further evaluated in a long-term test.

## 2. Materials and Methods

### 2.1. Membrane Preparation

PVDF (molecular weight of 534 kDa by GPC, Sigma Aldrich, MO, USA), dimethylacetamide (DMAC, Sigma Aldrich, MO, USA) and deionized water were used as a polymer, solvent and non-solvent, respectively to prepare phase inverted membrane. The polymer solution was prepared by dissolving 15 wt % of PVDF into DMAC. The mixtures were then stirred at 150 rpm and 60 °C [22] for at least 24 h to form a homogeneous dope solution. It was followed by degassing for overnight to release entrapped air bubbles.

The corrugated membrane preparation is illustrated in Figure 1. The corrugated membrane was prepared by employing a two-stage casting: (1) a corrugated layer and (2) a flat layer cast on the bottom side of the first one. To prepare the corrugated layer, the dope solution was first cast on non-woven support (NWS, Novatexx 24413, Freudenberg-filter, Weinheim, Germany) using a casting knife with a wet-casting thickness of 220 μm. A spacer (Sepa^®^ CF Feed Spacer, Sterlitech, Kent, WA, USA) as shown in Figure 2 was used as a template to form corrugations via imprinting the spacer into the wet casted film before immersing the casted film into a coagulation bath containing pure water (non-solvent) for overnight. The membrane was then dried in an oven at 35 °C for 6 h before it was used as a backing material to form the second layer. The flat layer was then formed in the bottom surface of the corrugated layer on the other side of the non-woven. The dope solution was cast using a casting knife with a wet-casting thickness of 220 μm and was immediately immersed in a coagulation bath containing non-solvent. The solidified film was then maintained in the coagulation bath overnight to ensure complete removal of the solvent followed by drying at room temperature. This second layer acts as a barrier to prevent defect as a result of the templating process in the processing step. The flat membrane was prepared through a method similar to one explained elsewhere [23].

### 2.2. Membrane Characterization

The membrane thickness was measured (at least at 5 different locations chosen randomly) using a micrometer (Mitutoyo, Takatsu-ku, Kawasaki, Japan). The porosity of each sample was measured gravimetrically using the dry-wet method. Fourier transform infrared spectrometer (FTIR) in the mode of attenuated total reflection (ATR) was used to study the chemical bonds of the developed membranes. Contact angle (CA) goniometer was used to measure the CA of deionized water on the membrane using a sessile drop of 7 μL. Multiple measurements (at least 5) were taken at different sample locations to ensure data accuracy. Differential scanning calorimeter (DSC) was used to evaluate the degree of crystallinity of the PVDF using thermal analysis apparatus (TA Instrument, New Castle, DE, USA). The samples were heated from 25 to 200 °C at the rate of 10 °C/min. Total crystallinity, $X_{c}$ of PVDF was calculated by using Equation (1): (1)$$X_{c}=\frac{\Delta H_{f}}{\Delta {H_{f}}^{*}}\times 100\%$$
where $\Delta {H_{f}}^{*}$ is the melting enthalpy for 100% crystalline PVDF which is 104.7 J/g, and $\Delta H_{f}$ the melting enthalpy of samples obtained from DSC.

### 2.3. Direct Contact Membrane Distillation Set-Up and Operational Parameters

An illustration of a lab-scale direct contact membrane distillation (DCMD) used in this study is shown in Figure 3. The flat and corrugated membranes with a dimension of 4 cm × 8.5 cm were used as DCMD testing samples. In the first test, synthetic seawater (35 g/L of NaCl in DI water) with a conductivity of 63,000 µS/cm was used as the feed. The second test was performed using synthetic brine (70 g/L of NaCl in DI water) corresponding to 127,000 µS/cm of conductivity. Tap water with a conductivity of 70 µS/cm was used as the cold liquid in the permeate side. The feed and the permeate temperatures were maintained at 65 and 25 °C, respectively throughout the experiment. An electronic balance was used to measure permeate overflow weight over time. The weight and conductivity were recorded every 5 min during the test to be used later for calculating permeate flux and salt rejection. All DCMD experiments were carried out using hot and cold solutions with a constant linear velocity of 2.2 cm/s using a pump. A hotplate (C-MAG HS7) with temperature controller was used to maintain the temperature of the feed while a continuous flow of tap water outside of the permeate tank was utilized to maintain the temperature of the permeate. The flux was determined based on the permeate overflow using a weighing balance. The flux performance was calculated based Equation (2)
(2)$$\mathrm{Permeate}\;\mathrm{Flux}=\frac{V}{A\;\Delta t}$$
where V is the volume of permeate collection in time (h), A (m^2^) is the membrane effective area which is 3.6 × 10^−3^ and ∆t (h) is the time taken for the test.

The conductivity of permeate was measured from time to time to study the salt rejection by using conductivity meter (Mettler Toledo Seven Excellence S470). After solving the salt and water balance to find the conductivity of the permeate, the salt rejection was calculated using Equation (3).
(3)$$\mathrm{Salt}\;\mathrm{rejection}=1-\frac{C_{\mathrm{permeate}}}{C_{\mathrm{feed}}}\times 100\%$$
where C_feed_ and C_permeate_ are the salt concentration in the feed and permeate respectively (g/L).

The flow regime of the membrane module also was determined using the Reynold number (R_e_) formula, represented by Equation (4)
(4)$$R_{e}=\frac{\rho \mathrm{VD}}{\mu }$$
where ρ is the density of the fluid, V is the fluid velocity, μ is the viscosity of the fluid and D is the dimension of the fluid. By using Equation (4), the Re of feed and cold side flow was 160 indicating that both flows were in the laminar regime.

## 3. Results and Discussion

### 3.1. Membrane Properties

#### 3.1.1. Pore Size and Distribution

Figure 4a clearly shows that the imprinting step leads to higher mean flow pore size for the corrugated PVDF membrane (1.3 µm) compared to the flat PVDF membrane (0.14 µm). This contradicts to the findings reported in Reference [17], where dual-layer casting depresses the mean flow pore size of the PVDF membrane. The large pore size of the current PVDF membrane can promote flux but might be associated with reduced wetting resistance. In the case of a flat membrane, the spike of distribution at size 0.11 µm indicates that an abundant number of pores at the corresponding size. The cross-section images of the corrugated membranes shown by Figure 4c confirms the formation of corrugation shape on the membrane surface while Figure 4b shows the cross section of the fabricated flat membrane.

#### 3.1.2. Contact Angle, Thickness and Porosity

Table 2 summarizes the properties of the developed PVDF membranes. It includes CA of the top and bottom surfaces of membranes, overall thickness, porosity and pore size. The CA of the prepared flat membrane is reported to be lower than the corrugated membrane. A higher CA obtained by corrugated membrane might be due to the uneven surface structure contributed by templating of net-spacer during fabrication [17]. High CA helps to improve wetting resistance in DCMD.

The employment of a strong non-solvent (water) during membrane fabrication resulted in the surface “flattening effect” that reduces the CA of the membrane top surface [22]. Flattening effect is a term that is used to describe the effect of drastic polymer movement from the top cast-film to the bulk of film due to the distinct difference between the surface (cast-film and air) and interfacial (cast-film and water) tensions. The long-chain nature of the PVDF polymer initiates the formation of flat and smooth film surface which reduces the surface CA [24]. The absence of surface structural features depresses the surface hydrophobicity properties as reported elsewhere [24,25,26,27,28]. Despite its low CA, some reports [17,29,30] have shown that to be applicable for MD, the surface CA of a membrane does not have to be hydrophobic (CA > 90).

On the other hand, the CA of the membrane bottom surface are all higher (109.5–113.4°) compared the top one (80.4°–95.4°). This can be justified by the different rate of liquid-liquid and liquid-solid demixing that occurred in the polymer matrix [23]. The rapid exchange of solvent and non-solvent quickly solidifies the top skin layer which then provides more time for liquid-solid demixing to take place at the bottom part of the film [31]. Under those circumstances, more micro-structures formed at the bottom, which increase the CA.

The thickness of the membranes is determined by the wet thickness of film during fabrication. Table 2 shows that the dry thickness of the corrugated membrane (175 µm) is higher than the flat membrane (155 µm). The difference of membrane thickness is because of the different number of casted film layers (Figure 1) and the extra height of the ‘hills’ of the corrugation [7,17]. Meanwhile, the porosity of the corrugated membrane (65%) was obviously higher than the flat membrane (41%) due to the larger pore size. The porosity values for both membranes were acceptable for MD application as reported elsewhere [32]. It is worth noting that high porosity is one of the desired properties for the MD membrane since it offers more space for vapor transportation from the feed to the permeate side [33].

#### 3.1.3. FTIR-ATR Spectroscopy

Figure 5 shows the FTIR-ATR spectra of the flat and corrugated membranes. Both membranes show identical peaks, but with different intensities at certain wavenumbers. There are strong absorption band at 720, 840 and 880 cm^−1^ which attributed to C–H rocking and C–F stretching vibration of PVDF. Other than that, the presence of absorption band at 1074–1400 cm^−1^ corresponds to the C–C bending while absorption band at 2850 and 2916 cm^−1^ might be related to C–H stretching, as described elsewhere [26,34]. A higher peak intensity for the flat membrane at 720 cm^−1^ indicates that the membrane consists of more α-phase compared to the corrugated membrane. The appearance of a peak at 840 cm^−1^ for both membranes is due to the presence of the β-phase crystalline structure [35].

#### 3.1.4. Differential Scanning Calorimetry

The heat of fusion and total crystallinity of the developed membranes are summarized in Table 3. The fusion heat could be obtained by integrating the area under the DSC curves. Figure 6 shows the DSC curve of the developed corrugated and flat membrane. The degree of crystallinity was calculated by determining the heat of fusion of the samples. Those values were compared to the heat of fusion of a perfect crystalline PVDF to obtain the degree of crystallinity, as summarized in Table 3. The finding shows that the crystallinity degree for the corrugated membrane is 41.1%, which is lower compared to 45% for the flat membrane. This might be due to the different rate of liquid-liquid and liquid-solid demixing.

The first peak at 129 °C indicates the melting of non-woven support. The presence of two obvious peaks at temperatures of 159 and 169 °C for the flat membrane, 160 and 175 °C for the corrugated membrane represents the melting points of PVDF polymer. The distinction between two peaks at these temperatures is because of the different crystalline phases that coexist in the membrane matrix [36,37]. The first peak of polymer melting point which is at ~160 °C demonstrates the imperfect crystalline region, while the melting of the perfect region occurs at the second peaks (169 and 175 °C). Imperfect crystallization is facilitated by rapid liquid-liquid demixing and solidification process. It can be postulated that the fast demixing, in combination with rapid solidification of the top film, restricts the expansion of the cast film to the upper direction for the corrugated membrane, while its effect on the flat membrane is minimal. Under such circumstances, rearrangement of the PVDF chain during crystallization is thus limited.

### 3.2. Performance of Fabricated PVDF Membrane

#### 3.2.1. Short-Term DCMD Performance

Figure 7 shows that for treating seawater as the feed, the corrugated membrane offers a significant advantage over the flat one only at a higher temperature (65 °C). While Figure 8 shows that for brine as the feed that more vulnerable from scaling, the corrugated membrane shows advantages on flux overall testing temperature. Figure 7 shows the results of flux performance and salt rejection of both the flat and corrugated membranes when synthetic seawater solution was employed as feed. The fluxes of both membranes are increased with increasing feed temperature. In general, the increase in feed temperature increases the partial pressure difference, enhancing the mass transfer rate across the membrane. The fluxes obtained by both membranes were almost the same when the feed inlet was set at 45 °C due to the limited driving force to make a noticeable difference. As the feed temperature was increased to 55 °C, both membranes showed better fluxes; highest fluxes were attained when the inlet temperature was 65 °C, recording 18.0 and 23.1 L/m^2^h for the flat and the corrugated membrane, respectively, with almost complete salt rejection. All membranes can be potentially considered for MD owing to excellent salt rejection (98 to 99%) achieved during DCMD short-term tests.

Despite having a thicker film layer that would increase mass transport resistance, the corrugated membrane shows 27% higher flux compared to the flat membrane. This might be attributed to its larger pore size and a higher percentage of porous structures that enhances vapor transportation. Besides, corrugated membrane consists of more imperfect crystalline structures (Figure 6) and less degree of total crystallinity might partially contribute to better membrane performance in MD since it provides more free-space for water vapor to pass through. In addition, the corrugated nature of the corrugated membrane increases the effective surface area and results in higher flux compared to the flat membrane. The presence of more β-phase on the surface of the corrugated membrane according to FTIR result (Figure 5) might contribute to the improvement of flux. Chain arrangement in β-phase possesses the maximum dipole moment [38], which attracts polar molecules such as water. However, the high hydrophobic nature of the membrane surface repels the water liquid while allowing only vapor to pass through the pores. This phenomenon improves mass transport while maintaining its wetting resistant.

Figure 8 shows the flux and salt rejection of developed during short-term DCMD testing using synthetic brine as feeds at varying temperatures. As expected, both membranes show great improvement in flux as the feed temperature was raised to 65 °C. However, each shows noticeable flux decrement (27.7 and 36.2% for the flat and corrugated membrane, respectively) when synthetic brine was used as feed instead of synthetic seawater. The prominent reduction in flux is due to higher salinity of brine solution that reduces the feed vapor pressure in MD [39,40]. For the salt rejection pattern when the brine solution was applied as feed, both membranes show steady performance above 99%, except for the flat membrane at 45 °C (93%). This might be due to total pore wetting which allows the formation of salt bridging in the membrane pores and consequently result in poor rejection. When compared with the literature data [41,42,43,44], the membrane flux of this work is considered rather low possibly due to low applied cross-flow velocity, limited by the experimental set-up.

#### 3.2.2. Long-Term Performance (Wetting Resistance Test)

Long-term permeability was conducted to study the performance stability of the developed membranes. Figure 9 shows the trends of flux and salt rejection of flat and corrugated membrane for a 50-h operation by applying a brine solution as the feed. The temperature of permeate and feed was controlled at 25 and 65 °C, respectively. Both flux and salt rejection of FM are quite stable for the entire testing period with an average flux of 6.1 L/m^2^h and salt rejection of 99.1%. Unfortunately, the corrugated membrane experiences decreasing flux and salt rejection starting from the 25th hours of operation. The initial flux of the corrugated membrane was 14.6 L/m^2^h but gradually decreased to 7.7 L/m^2^h after 50-h operation. Similarly, a salt rejection started at 99.7%, but reduced to 84.5% at the end of the operation.

In comparison to a prior study [17], the currently developed corrugated membrane has significantly higher initial flux but seems to suffer from severe wetting and scaling. The main factor causing wetting and/or scaling is due to low applied cross-flow velocity during DCMD [33] and largely from its large pore size (1.3 µm). Low cross-flow velocity is a disadvantage for corrugated membrane since it is unable to produce turbulence flow regimes and increases the tendency of scaling and fouling. The future challenge is to develop a corrugated membrane with a slightly lower pore size that can offer not only higher flux but also wetting and scaling resistant.

## 4. Conclusions

A PVDF-based membrane with the corrugated surface was successfully developed in this study. The corrugation on the membrane surface improves the membrane features and increases the pore size of its active layer. Consequently, it manages to improve the membrane flux (up to 23.1 L/m^2^h) compared to a reference flat membrane by 27% during both short (20 min) and long (50 h) term tests (15 vs. 8 L/m^2^h). The flux increment is ascribed to its higher effective filtration area, larger pore size and, when treating brine, due to its positive role in improving local mixing thus minimize scaling. However, the corrugated membrane suffers from wetting and scaling in long term test that diminishes its flux performance. The performance of the corrugated membrane, however, was still better than the flat membrane. The decreased flux obtained corrugated membrane was due to its large pore size coupled with a low cross-flow velocity applied during DCMD test. This membrane is attractive for desalinating feed containing high salts, such as for RO brine distillation. In this case, the can be coupled with an existing desalination plant to achieve higher water recovery.

## Figures and Tables

**Figure 1 polymers-11-00865-f001:**
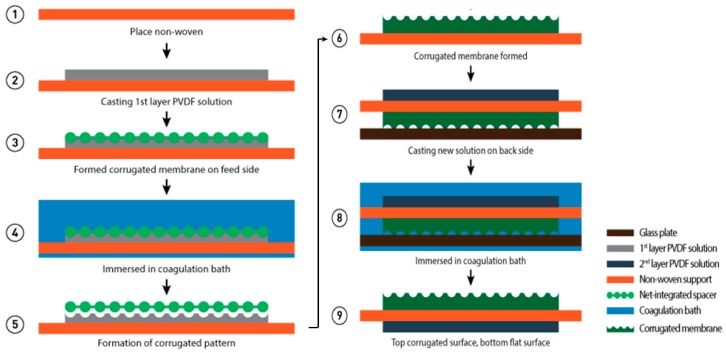
Illustration of the preparation method of the corrugated membrane. (1) Non-woven support, (2) casting of 1st layer of polyvinylidene difluoride (PVDF) solution on the non-woven support to form a thin polymer solution film, (3) formation of the corrugated membrane on feed side, (4) immerse the corrugated membrane in a coagulation bath containing water as non-solvent, (5) delamination of the spacer from solidified corrugated membrane, (6) formation of the solidified corrugated membrane, (7) casting of the 2nd layer of PVDF film on the back side of the non-woven, (8) immersed the membrane film in the coagulation bath, and (9) corrugated solidified membrane was formed on the top and bottom side is flat.

**Figure 2 polymers-11-00865-f002:**
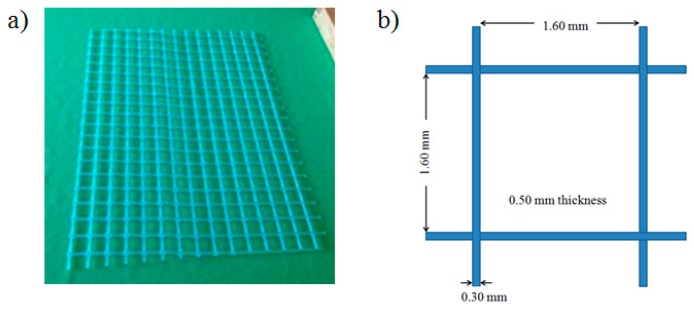
(**a**) Image of the imprinted spacer and (**b**) its dimensions.

**Figure 3 polymers-11-00865-f003:**
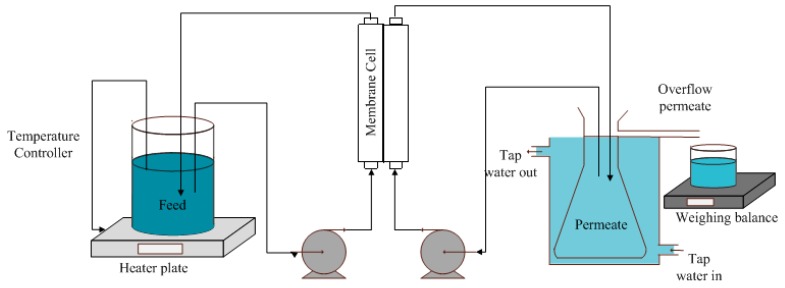
Illustration of experimental direct contact membrane distillation set-up.

**Figure 4 polymers-11-00865-f004:**
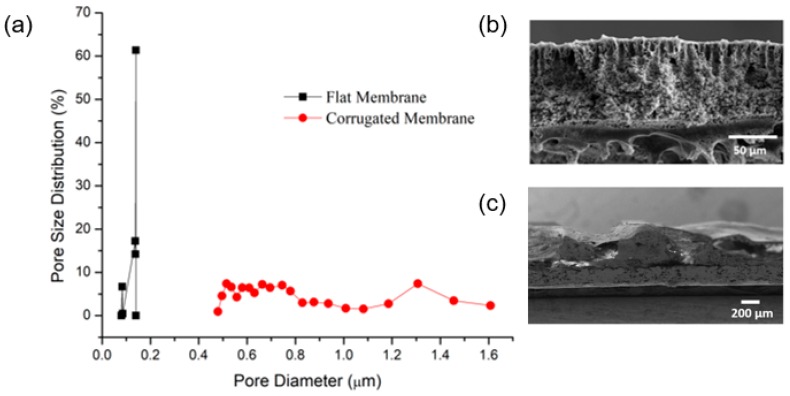
(**a**) Pore size and distribution of the membrane samples. SEM cross-sectional image of (**b**) flat and (**c**) corrugated membranes. Notice the different magnifications of the SEM images in (**b**,**c**).

**Figure 5 polymers-11-00865-f005:**
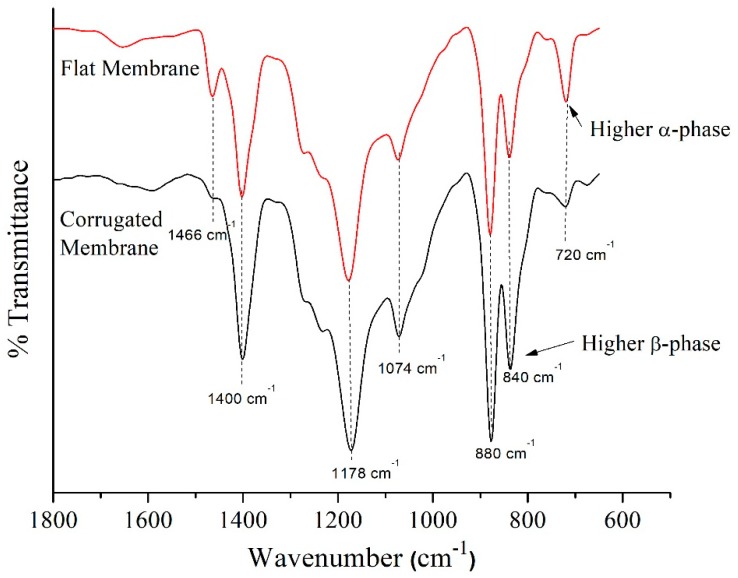
The FTIR spectra of the developed flat and corrugated PVDF membranes.

**Figure 6 polymers-11-00865-f006:**
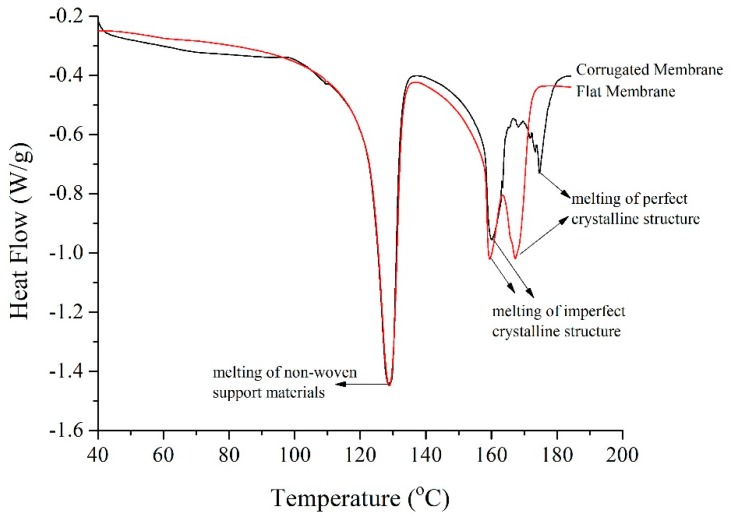
Differential scanning calorimeter curve of the developed membranes.

**Figure 7 polymers-11-00865-f007:**
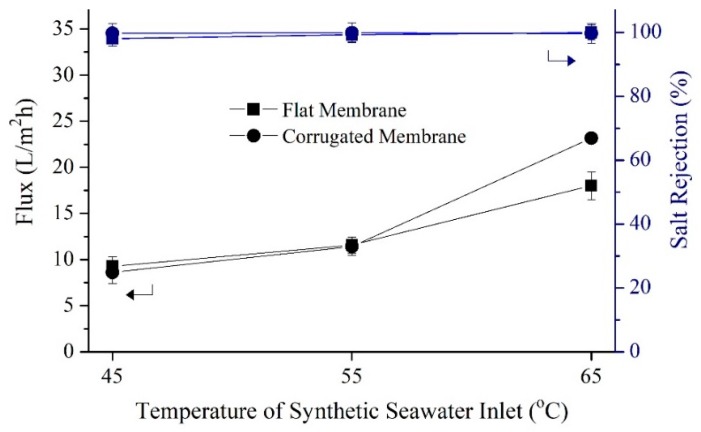
Short-term performance of flat and corrugated PVDF membranes in treating synthetic seawater (35 g/L of NaCl in deionized water) solution at different feed temperatures. The cold stream was maintained at 25 °C and both hot and cold streams were pumped at a constant linear velocity of 2.2 cm/s.

**Figure 8 polymers-11-00865-f008:**
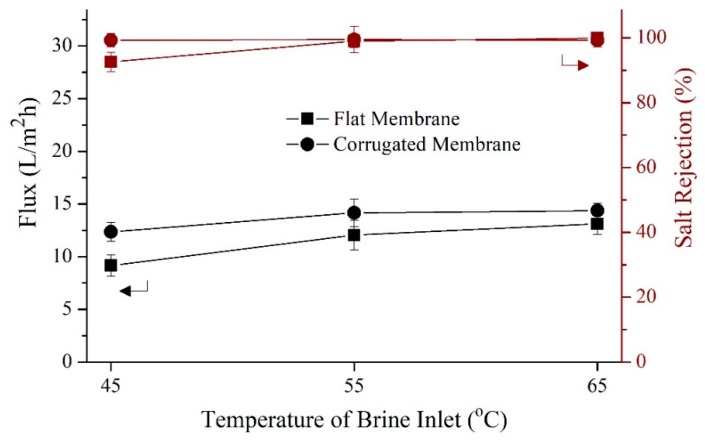
Performance of flat and corrugated membranes in treating brine solution (70 g/L of NaCl in DI water) at different feed temperatures. The cold stream was maintained at 25 °C. both cold and hot streams were pumped at a linear velocity of 2.2 cm/s.

**Figure 9 polymers-11-00865-f009:**
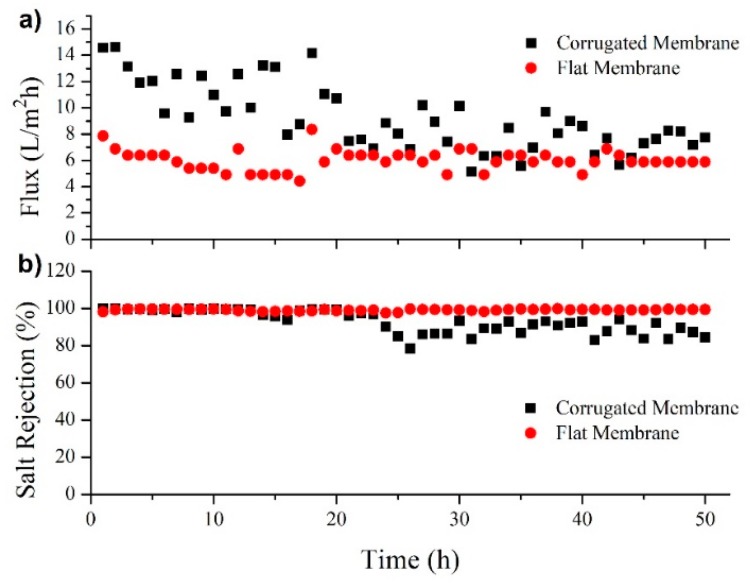
Long-term membrane distillation performance in term of flux (**a**) and salt rejection (**b**) for the treatment of brine feed (70 wt % of NaCl in DI water). The temperatures of the hot and cold sides were maintained at 65 and 25 °C respectively. Both streams were pumped at a linear velocity of 2.2 cm/s.

**Table 1 polymers-11-00865-t001:** Recent report on a corrugated membrane and corrugated membrane plate.

Application	Membrane Material	Method of Corrugation Formation	Year	Ref
Water-oil-emulsion microfiltration	PTFE	Mechanical pressing between metal dies at 120 °C	2000	[20]
Membrane distillation	PVDF	Imprinting method	2015	[17]
Membrane bioreactor	PVDF	Imprinting method	2015	[7]
Forward osmosis (simulation)	N/A	N/A	2019	[21]

**Table 2 polymers-11-00865-t002:** The properties of the developed PVDF membranes.

Membrane	Contact Angle (°)		Thickness (µm)	Porosity (%)	Pore Size (µm)
	Top Surface	Bottom Surface			
Corrugated	94.5 ± 8.8	109.5 ± 2.4	172 ± 5.6	65 ± 7.1	1.30
Flat	80.4 ± 4.7	113.4 ± 3.0	155 ± 4.8	41 ± 5.0	0.14

**Table 3 polymers-11-00865-t003:** Degree of crystallinity of the developed membranes based on Differential scanning calorimeter testing.

Membrane	Heat of Fusion, ΔH_f_ (J/g)	Total Crystallinity, X_c_ (%)
Corrugated	43.36	41.4
Flat	47.12	45.0

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
