# Peer review of "Development of A Novel Corrugated Polyvinylidene difluoride Membrane via Improved Imprinting Technique for Membrane Distillation"

_polymers, 2019, doi:10.3390/polym11050865_

Round 1
Reviewer 1 Report
This paper focuses on addressing those limitations by developing a novel corrugated polyvinylidene difluoride (PVDF) membrane via an improved imprinting technique for MD. Corrugations on the membrane surface are designed to offer effective surface area and at the same time act as turbulence promoter to induce hydrodynamic by reducing temperature polarization. My review comments are as follows:
Ref. 1, please change the reference format.
Some latest references about the different treatment methods for wastewater should be mentioned. Please see the related work (10.1016/j.jenvman.2018.08.069). What are the importance of the various components in the wastewater?
Why did you choose the membrane material of PVDF?
Line 134, please delete 'Eq.'.
In Fig. 3, why does the flat membrane show a huge difference in pore size flow?
Please improve the figures. There are a borderline in most figures.
In Figs. 6 and 7, both membranes showed the similar results of salt rejection and even the flux. What is the main impact of the comparison?
What is the industrial potentials of your membranes?
Please double check the reference format in the revision.
Author Response
Dear Reviewer,
Thank you for your constructive feed back on our manuscript. Please find our detailed response in the attached file.
Please see the attachment.
Thanks

Reviewer 2 Report
Major Revision
The author has demonstrated the feasibility of utilizing novel corrugated PVDF membrane fabricated via imprinting technique for MD application. Actually, this is good based topic that may arise some researcher's interests studying membrane fabrication and MD. Although an extensive research output has been provided but there are some serious flaws that were mentioned below. Kindly modify the manuscript while considering the following points:
1. To make this research article more attractive, the author must include the previous research outputs of corrugated membranes (with conditions such as polymer used, technique used for membrane fabrication etc.) by comparing the present one in tabular form in order to show the viability of the present material/technique.
2. Very importantly, mathematical modeling such as Reynold’s number is missing from the manuscript. The author must indicate whether the flow was laminar or turbulent by indicating the Reynold’s number.
3. Next, while dealing with membrane subject, the author must discuss these following properties. These general characteristics are missing from the present manuscript (Please provide following data whatever possible). .
(a) Morphological structure by SEM
(b) Cross sectional view by SEM
(c) Surface roughness by AFM
(d) Dynamic mechanical strength of the membrane
(e) If possible, prototype of the fabricated material (as the author claimed it to be novel).
4. The contact angle is really low (94° for top layer) compared to other commercial MD membranes. So, the consistency in water flux as well as rejection % is really doubtful. As after a certain period, the membrane may lose hydrophobicity and that may influence the performance in MD process. The author should come out with high scientific discussion to defend this question.
Minor Revision
1. The quality of conclusion is really below par standard. Kindly put some effort while drafting the conclusion. Try to add more scientific discussion to make it more attractive for readers.
2. Even, materials and methods section must be improved. The mathematical formula(s) for water flux, salt rejection and Reynold’s number must be included in the revised version.
3. Kindly cite few papers from “Polymers: MDPI Journal”

Author Response
Dear Reviewer,
Thank you for your constructed comments on our manuscript. We believe that the manuscript is now much improved in quality after addressing your comments in the revised version.
Please see the attachment.
Thanks

Round 2
Reviewer 1 Report
This paper could be accepted now.
Reviewer 2 Report
The manuscript can be accepted in the present format.